# A Two-Step Deep Learning Approach for Abdominal Organ Segmentation

Jianwei Gao[0000−0002−6358−4117], Juan Xu, Honggao Fei, and Dazhu Liang

Digital Health China Technologies Co., LTD, Beijing, China
{gaojw,xujuan,feihg}@dchealth.com

**Abstract.** Accurate delineation and analysis of anatomical structures within medical images are essential in various clinical applications, with medical image segmentation playing a key role. In the context of abdominal imaging, the precise segmentation of organs like the liver, spleen, and kidneys holds significant importance for tasks such as diagnosis, treatment planning, and surgical interventions. However, achieving precise and efficient segmentation of abdominal organs poses significant challenges due to the variability in organ shape, size, and appearance across different patients and imaging modalities. The MICCAI FLARE23 segmentation paper presents a solution to the challenging problem of segmenting 13 organs and tumor from CT scans, provided 2200 CT scans with partial labels and 1800 CT scans without labels, while balancing model performance and resource consumption. To address these challenges, the paper proposes a two-step segmentation approach that combines organ segmentation and tumor segmentation, which are both accomplished with nnU-Net model. We also crop some top and bottom slices for faster process.

**Keywords:** Abdominal organ segmentation · Supervised Learning · nnUnet.

## 1 Introduction

Medical image segmentation plays a pivotal role in various clinical applications, enabling the accurate delineation and analysis of anatomical structures within medical images. However, achieving precise and efficient segmentation of abdominal organs poses significant challenges, because it typically requires a large amount of labeled data to train an accurate model, while manually annotating organs from CT scans is a time-consuming and labor-intensive process, furthermore, abdominal organs may have complex morphological structures and heterogeneous lesions, which segmentation a more difficult task.

In recent years, deep learning became the mainstream method for medical image analysis, demonstrating remarkable capabilities in automated organ segmentation tasks. [9] Specifically, the nnU-Net model [7] has emerged as a powerful framework for achieving state-of-the-art results in medical image segmentation. nnU-Net combines the strengths of the U-Net architecture with advancements in neural network design and training strategies, allowing for improved accuracy

and robustness. Semi-supervised segmentation [15] is a type of segmentation where the training set consists of both labeled and unlabeled data. The goal is to assign pseudo-labels to the pixels of unlabeled images. This approach is useful when obtaining labeled data is expensive or time-consuming, which is perfect for this challenge.

Because there are no full 14 classes labeled data but 13 classes organ segmentation labeled data, in this paper,we break it down into two tasks: organ segmentation and tumour segmentation. Therefor, we propose an approach which involved training two nnUnet model with labeled data, which are used to segment organs and tumours respectively. A post-process is used to merge two Deep Learning results when inferencing.

## 2      Method

### 2.1    Preprocessing

We use several pre-processing strategies as follows.

- Data choose and preprocessing
  We choose train data with full 13 organ label and data with tumor label, thus get 222 data for organ segment and 735 for tumor segment. then we split them by 8:2 ratio for train and validation.
- Cropping strategy
  We use the CT scans as the data source to generate the bounding box of foreground, and then crop only the foreground object of the images.
- Resampling method for anisotropic data
  We resample the original data to unify the voxel spacing into $[1.0, 1.0, 1.0]$.
- Intensity normalization method
  We collect intensity values from the foreground classes (all but the background and ignore) from all training cases, compute the mean, standard deviation as well as the 0.5 and 99.5 percentile of the values. Then clip to the percentiles, followed by subtraction of the mean and division with the standard deviation. The normalization that is applied is the same for each training case (for this input channel).

### 2.2    Deep Network

Figure 1 illustrates the applied 3D nnU-Net [7], where a 3D U-Net architecture is adopted. We use the leaky ReLU function with a negative slope of 0.01 as the activation function. Our first 3D nnU-Net has 14 out channels, corresponding to the background and 13 organs, while our second 3D nnU-Net has 2 out channels, corresponding to the background and the tumor. In this case, only data with 13 organ label and data with tumor label are used, the others is abandon.Unlabeled images were not used.

We use the sum of Dice loss (after applying a softmax function) and Cross Entropy Loss as the loss function, because it's a popular choice for loss fuction and have been proven to be robust in various medical image segmentation tasks.

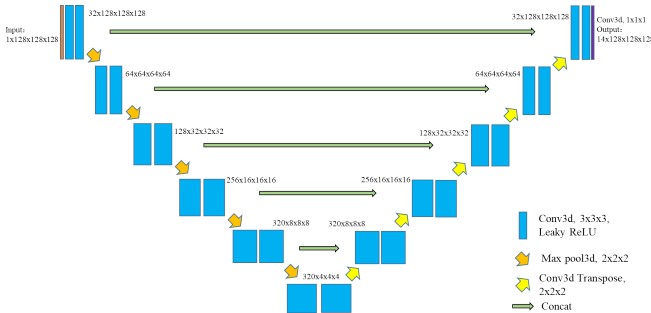

**Fig. 1.** Our 3D U-Net architecture

When predicting a single image with the trained segmentation model, we first resample it to a voxel spacing of [1.0, 1.0, 1.0], as we did during training, and try to predict. In order to improving inference speed and reducing resource consumption, we crop top and bottom slices of the data in predicting, only keep only the middle 50 slices.

### 2.3   Post-processing

During model prediction, we select the label (from 0 to 13) corresponding to the largest of the 13 outputs for each voxel, after that, we combine predictions of two model to get the final result. when one pixel is both predicted as tumor and organ, it will be considered as tumor. finally, we add full 0 array as top and bottom slices

## 3   Experiments

### 3.1   Dataset and evaluation measures

The FLARE 2023 challenge is an extension of the FLARE 2021-2022 [11][12], aiming to aim to promote the development of foundation models in abdominal disease analysis. The segmentation targets cover 13 organs and various abdominal lesions. The training dataset is curated from more than 30 medical centers under the license permission, including TCIA [2], LiTS [1], MSD [14], KiTS [5,6], autoPET [4,3], TotalSegmentator [16], and AbdomenCT-1K [13]. The training set includes 4000 abdomen CT scans where 2200 CT scans with partial labels and 1800 CT scans without labels. The validation and testing sets include 100 and 400 CT scans, respectively, which cover various abdominal cancer types, such as liver cancer, kidney cancer, pancreas cancer, colon cancer, gastric cancer, and so on. The organ annotation process used ITK-SNAP [17], nnU-Net [8], and MedSAM [10].

The evaluation metrics encompass two accuracy measures—Dice Similarity Coefficient (DSC) and Normalized Surface Dice (NSD)—alongside two efficiency

measures—running time and area under the GPU memory-time curve. These metrics collectively contribute to the ranking computation. Furthermore, the running time and GPU memory consumption are considered within tolerances of 15 seconds and 4 GB, respectively.

## 3.2   Implementation details

**Environment settings** The development environments and requirements are presented in Table 1.

**Table 1.** Development environments and requirements.

| | |
|---|---|
| Windows/Ubuntu version | Ubuntu 20.04.4 LTS |
| CPU | Intel(R) Xeon(R) Gold 5218R CPU @ 2.10GHz |
| RAM | 128G |
| GPU (number and type) | NVIDIA Tesla T4 (16G) |
| CUDA version | 11.6 |
| Programming language | Python 3.9 |
| Deep learning framework | Pytorch (Torch 1.13.1, torchvision 0.14.1) |
| Specific dependencies | numpy 1.25.2, SimpleITK 2.2.1, nnunetv2 2.1 nibabel 5.1.0 |

**Training protocols** As described below.

Random flipping strategy (only for initial training stage): each image has a 20% probability of flipping along the x-axis and a 20% probability of flipping along the y-axis.

Random Gaussian smooth (only for initial training stage): each image has a 10% probability of being Gaussian smoothed with sigma in (0.5, 1.15) for every spatial dimension.

Random Gaussian noise (only for initial training stage): each image has a 20% probability of being added with Gaussian noise with mean in (0, 0.5) and standard deviation in (0, 1).

Random intensity change (only for initial training stage): each image has a 10% probability of changing intensity with gamma in (0.5, 2.5).

Random intensity shift (only for initial training stage): each image has a 10% probability of shifting intensity with offsets in (0, 0.3).

Patch sampling strategy: 2 patches of size $[128, 128, 128]$ are randomly cropped from each image. The center of each patch has 50% probability in the foreground and 50% probability in the background.

As described above,only data with 13 organ label and data with tumor label are used, the others is abandon. Unlabeled images were not used.

Some details of the initial training stage and the fine-tuning stage are shown in Table 2 and Table 3 respectively.

**Table 2.** Training protocols (initial training stage).

| | |
|---|---|
| Network initialization | "he" normal initialization |
| Batch size | 2 |
| Patch size | 128×128×128 |
| Total epochs | 1000 |
| Optimizer | Adam |
| Initial learning rate (lr) | 0.0001 |
| Lr decay schedule | initial learning rate$\times(1-epoch/500)^{0.9}$ |
| Training time | 20 hours |
| Loss function | the sum of dice loss and cross entropy loss |
| Number of model parameters | 31.42M |

**Table 3.** Training protocols (fine-tuning stage).

| | |
|---|---|
| Network initialization | model after initial training |
| Batch size | 2 |
| Patch size | 128×128×128 |
| Total epochs | 40 |
| Optimizer | Adam |
| Initial learning rate (lr) | 0.00005 |
| Lr decay schedule | initial learning rate$\times(1-epoch/500)^{0.9}$ |
| Training time | 39 hours |
| Loss function | the sum of dice loss and cross entropy loss |
| Number of model parameters | 31.42M |

## 4    Results and discussion

### 4.1    Quantitative results on validation set

DSC and NSD results on validation set are shown in Table 4. It can be seen from the table that Aorta and LK have best proformance, while others has worst proformance.A possible reason of it is that Aorta and LK is larger organ and more likely in the center,therefor not be croped by preprocessing.

**Table 4.** Results on validation set.

| Target | Public Validation | | Online Validation | | Testing | |
|--------|-------------------|-----|-------------------|-----|---------|---------|
|  | DSC(%) | NSD(%) | DSC(%) | NSD(%) | DSC(%) | NSD (%) |
| Liver | 0.600± 0.600 | 0.000± 0.000 | 5.500 | 4.400 | | |
| RK | 1.500± 1.500 | 1.110± 1.110 | 5.500 | 8.900 | | |
| Spleen | 0.000± 0.000 | 0.000± 0.000 | 10.000 | 10.000 | | |
| Pancreas | 0.440± 0.440 | 0.540± 0.540 | 5.100 | 7.500 | | |
| Aorta | 8.860± 8.860 | 9.570± 9.570 | 6.870 | 7.460 | | |
| IVC | 0.980± 0.980 | 0.820± 0.820 | 14.500 | 14.000 | | |
| RAG | 4.000± 4.000 | 4.000± 4.000 | 2.000 | 2.000 | | |
| LAG | 2.000± 2.000 | 2.000± 2.000 | 1.000 | 1.000 | | |
| Gallbladder | 8.000± 8.000 | 8.000± 8.000 | 10.000 | 10.000 | | |
| Esophagus | 0.000± 0.000 | 0.000± 0.000 | 0.0000 | 0.0000 | | |
| Stomach | 0.000± 0.000 | 0.000± 0.000 | 0.0000 | 0.0000 | | |
| Duodenum | 5.500± 5.500 | 9.400± 9.400 | 0.280 | 0.470 | | |
| LK | 11.930± 11.930 | 11.480± 11.480 | 12.210 | 11.090 | | |
| Tumor | 0.000± 0.000 | 0.000± 0.000 | 0.000 | 0.000 | | |

**Table 5.** Quantitative evaluation of segmentation efficiency in terms of the running them and GPU memory consumption. Total GPU denotes the area under GPU Memory-Time curve.

| Case ID | Image Size | Running Time (s) | Max GPU (MB) | Total GPU (MB) |
|---------|-----------|------------------|--------------|----------------|
| 0001 | (512, 512, 55) | 133.13 | 3746 | 30428 |
| 0051 | (512, 512, 100) | 143.65 | 3806 | 14327 |
| 0017 | (512, 512, 150) | 121.93 | 3590 | 24825 |
| 0019 | (512, 512, 215) | 85.79 | 3323 | 14863 |
| 0099 | (512, 512, 334) | 73.7 | 3374 | 11619 |
| 0063 | (512, 512, 448) | 72.57 | 3370 | 10082 |
| 0048 | (512, 512, 499) | 70.87 | 3316 | 10466 |
| 0029 | (512, 512, 554) | 74.38 | 3382 | 11035 |

### 4.2   Qualitative results on validation set

Two examples of good segmentation are shown in Figure  2 and two examples of bad segmentation are shown in Figure  3. Visualization is achieved with ITK-SNAP [18] version 3.6.0.

From the perspective of images, some potential reasons for the bad-segmentation cases are listed below.

(1) The size of the case is very large, so we have to reduce the size of the case by cutting top and bottom slice to process it in 60 second.

(2) The case is not clear, distorted, or skewed.

(3) There are rare structures in the case that are not in the training set.

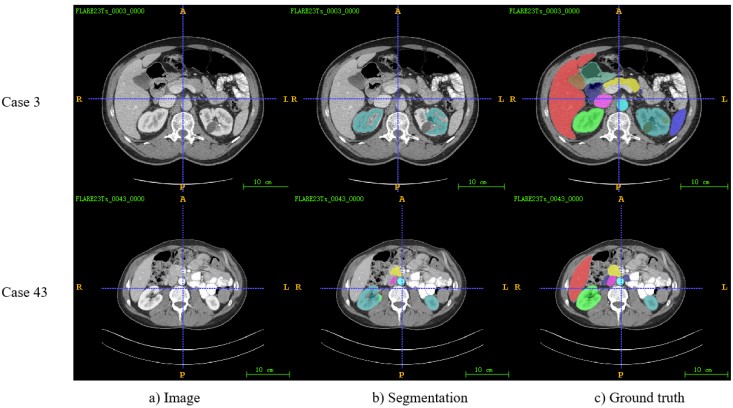

**Fig. 2.** Good segmentation examples

### 4.3   Segmentation efficiency results on validation set

Table 5 show the efficiency results on 8 validation sets.Due to the crop of top and bottom slices, the data which has larger thrid-dimension also have fast running times.

### 4.4   Results on final testing set

### 4.5   Limitation and future work

In terms of model accuracy, first, we does not use pseudo-labels for unlabeled image at present. In the future, we are going to use pseudo-labels for unlabeled image. Second, we consider using some post-processing methods, such as largest connected component extraction, hole filling, open operation and closed operation, which are not used at present.

to reduce the time consumption, we simply cut top and bottom slices, which caused a large loss of accuracy. To deal with it, we consider using some optimization methods to improve the running speed of the model in the future.

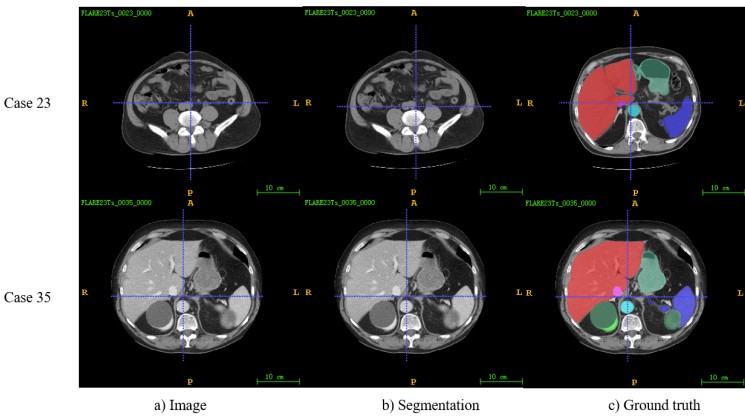

a) Image                    b) Segmentation                    c) Ground truth

**Fig. 3.** Bad segmentation examples

## 5    Conclusion

In this paper, we have explored the application of the nnU-Net model for Flare23 abdominal organ segmentation. Due to the limitation of time, we could not Leverage the power of deep learning and the architectural advancements of nnU-Net, but we will explore better deep learning methods in the future.

**Acknowledgements** The authors of this paper declare that the segmentation method they implemented for participation in the FLARE 2023 challenge has not used any pre-trained models nor additional datasets other than those provided by the organizers. The proposed solution is fully automatic without any manual intervention.

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

**Table 6.** Checklist Table. Please fill out this checklist table in the answer column.

| Requirements | Answer |
| --- | --- |
| A meaningful title | Yes |
| The number of authors ($\leq$6) | 4 |
| Author affiliations and ORCID | Yes |
| Corresponding author email is presented | No |
| Validation scores are presented in the abstract | No |
| Introduction includes at least three parts: background, related work, and motivation | Yes |
| A pipeline/network figure is provided | Figure 1 |
| Pre-processing | Page 2 |
| Strategies to use the partial label | Page 2 |
| Strategies to use the unlabeled images. | Page 2 |
| Strategies to improve model inference | Page 3 |
| Post-processing | Page 3 |
| Dataset and evaluation metric section is presented | Page 3 |
| Environment setting table is provided | Table 1 |
| Training protocol table is provided | Table 2 |
| Ablation study | Page 7 |
| Efficiency evaluation results are provided | Table 5 |
| Visualized segmentation example is provided | Figure 2/3 |
| Limitation and future work are presented | Yes |
| Reference format is consistent. | Yes |