# OpenReview forum: "A Two-Step Deep Learning Approach for Abdominal Organ Segmentation"
_MICCAI.org/2023/FLARE — Submitted to FLARE 2023_

### Official Review · Reviewer_J3Nk · 2023-09-27
**the author uses a two-step segmentation approach for organ and tumor segmentation**

**Rating:** 4
**Confidence:** 4

**Review:**

Pros：
the author uses a two-step segmentation approach for organ and tumor segmentation.

Cons:
Please refer to the official template to revise the paper.Such as In the abstrace section, the author should introduce  the validation performance.

---

### Official Review · Reviewer_VXxZ · 2023-10-04
**Review for "A Two-Step Deep Learning Approach for Abdominal Organ Segmentation"**

**Rating:** 4
**Confidence:** 4

**Review:**

The article explored the application of the nnU-Net model for FLARE23 abdominal organ segmentation.

Cons:
1. The author didn't introduce the performance of their proposed method in the abstract.
2. The author didn't include the related work/state-of-the-art methods on semi-supervised/partial-label segmentation in Introduction section.
3. Not all authors' ORCIDs have been provided.
4. In the second section, no detailed explanation of the method was provided.
5. No ablation studies conducted.

---

> ### Comment · Reviewer_VXxZ · 2023-11-30
> **2nd round Review**
>
> The author did not respond to the previous review comments or make any modifications to the paper.
>
> In terms of completeness, there are still significant issues with this article at present.

---

### Public Comment · ~PENGJU_LYU1 · 2023-11-26
**add test results**

add test results in table 4

---

### Decision · Program_Chairs · 2023-10-24

**Decision:**

Reject

**Comment:**

The authors didn't make responses to the valuable review comments.